# Washable and Flexible Screen-Printed Ag/AgCl Electrode on Textiles for ECG Monitoring

**DOI:** 10.3390/polym15183665

**Published:** 2023-09-06

**Authors:** Huating Tu, Xiaoou Li, Xiangde Lin, Chenhong Lang, Yang Gao

**Affiliations:** 1College of Medical Instruments, Shanghai University of Medicine & Health Sciences, Shanghai 201318, China; tuht@sumhs.edu.cn (H.T.); lixo@sumhs.edu.cn (X.L.); linxd@sumhs.edu.cn (X.L.); 2Key Laboratory of Intelligent Textile and Flexible Interconnection of Zhejiang Province, Zhejiang Sci-Tech University, Hangzhou 310018, China; 3School of Mechanical and Power Engineering, East China University of Science and Technology, Shanghai 200237, China; 4State Key Laboratory of New Textile Materials and Advanced Processing Technologies, Wuhan Textile University, Wuhan 430200, China

**Keywords:** textiles electrode, screen printing, Ag/AgCl conductive ink, electrocardiography

## Abstract

Electrocardiogram (ECG) electrodes are important sensors for detecting heart disease whose performance determines the validity and accuracy of the collected original ECG signals. Due to the large drawbacks (e.g., allergy, shelf life) of traditional commercial gel electrodes, textile electrodes receive widespread attention for their excellent comfortability and breathability. This work demonstrated a dry electrode for ECG monitoring fabricated by screen printing silver/silver chloride (Ag/AgCl) conductive ink on ordinary polyester fabric. The results show that the screen-printed textile electrodes have good and stable electrical and electrochemical properties and excellent ECG signal acquisition performance. Furthermore, the resistance of the screen-printed textile electrode is maintained within 0.5 Ω/cm after 5000 bending cycles or 20 washing and drying cycles, exhibiting excellent flexibility and durability. This research provides favorable support for the design and preparation of flexible and wearable electrophysiological sensing platforms.

## 1. Introduction

Nowadays, cardiovascular disease has become one of the major diseases that jeopardize human life and health globally. Continuous ECG monitoring with cardiac electrodes is one of the most effective means to effectively prevent cardiovascular diseases [1]. The cardiac electrode undergoes redox reactions to convert the weak ionic potential on the surface of the skin into a flow of electrons and requires both excellent electricity and stable electrochemical properties [2].

In medical testing or healthcare, the disposable Ag/AgCl glue electrodes are widely used due to their excellent electrical conductivity, low equilibrium potential and impedance, and minor motion artifacts. However, the weaknesses of their shelf life, their impermeability, and their skin allergy limit their wide usage for long-term monitoring [3]. Therefore, scientists have been attempting to create dry electrodes [4,5] since they are more appropriate for long-term monitoring and comfortable for daily wear. Actually, all those issues can be resolved by textile electrodes, which are based on textile structural materials combined with metal deposition technology.

In general, the textile electrodes are prepared in two ways as follows [6]: (1) weave metal-coated yarn into conductive fabrics with certain weaving processes [7], such as weaving [8], knitting [9], embroidery [10], and so on; and (2) deposit conductive materials (e.g., metals [11,12,13], graphene [14,15], polypyrrole [16], and poly(3, 4-ethylenedioxythiophene):poly(styrene sulfonate) (PEDOT:PSS) [17,18]) on the surface of the fabric using chemical methods or printing techniques. Ag/AgCl has proven to be one of the ideal materials for the electrode with low frequency recordings and exhibits very low offset voltage and baseline drift [19]. Moy, et al. [20] printed Ag ink and then chlorinated it with NaClO to fabricate the Ag/AgCl reference electrode, and the preparation processes are relatively complex. Kurniawan et al. [19,21] fabricated the sensor layer of the electrodes with conductive Ag/AgCl ink, but the complementary metal oxide semiconductor (CMOS) equipment and procedures were required. That is to say, the above processes increase the difficulty and cost of preparation for textile electrodes.

Screen printing offers a feasible, rapid-prototyping, and low cost way of producing functional wearable electronics, such as sensors [22,23,24], antennas [25,26], energy production and storage [27], and so on. A lot of researchers have screen printed conductive inks such as Ag [11] and Ag/AgCl [13] on ordinary fabric to fabricate textile electrodes, and the results have demonstrated that the capability of ECG signal acquisition is comparable to the conventional gel electrodes. However, the above screen-printed textile electrodes focus on the resistance or impedance evaluation while ignoring the electrochemical properties, which are very important for the performance of electrodes [28]. 

On the other hand, washability and durability are two major obstacles hindering the development of the wearable e-textile [28,29]. For flexible printed devices, the conductive coating of printed circuits may be damaged by washing or external mechanical action, causing discontinuities in the conductive pathway and thus affecting the service life of the device.

The above printed textile electrodes lack durability evaluation after bending or washing cycles [10,12,13], which are very important for the performance of electrodes.

In this work, a dry electrode for ECG monitoring was fabricated by screen printing Ag/AgCl mixed ink on ordinary polyester fabric. The preparation process was one-step printing, and the silver nanoparticles provided excellent electrical properties while the silver chloride stable ensured stable electrochemical performance. Moreover, a series of measurements and characterizations were performed to comprehensively assess the performance of the textile electrode, including surface topography, electrical and electrochemical characteristics, durability, and ECG signal acquisition. As illustrated in Figure 1, firstly, the Ag/AgCl conductive ink was deposited on the flexible fabric to form a specific electrode structure utilizing the screen-printing technique. Then, the conductive channel was created by drying the textile electrodes at a specific temperature to volatilize the solvent. Finally, the electrophysiological signals were collected by fixing the textile electrodes on the skin using the principle of redox reactions.

Herein, we aim to prepare textile ECG electrodes with stable performance using simple screen-printing methods and comprehensively assess the performances of electrical, electrochemical, durability, and signal acquisition capabilities. It will throw some light on the development of flexible and wearable sensors for monitoring electrophysiological signals.

## 2. Materials and Methods

### 2.1. Materials and Fabrication

The substrate of the textile electrode is made of polyester with a thickness of 0.093 mm. The fabric is a satin structure with a flat and smooth surface for screen printing. The linear density of the warp yarn is 256D, and of the weft yarn, it is 869D. The warp and weft densities of the satin woven fabric are 350 count/10 cm and 300 count/10 cm, respectively. All the above parameters of the fabric were conducted in the laboratory with constant temperature and humidity (22–23 °C, 63–65% RH).

The selected conductive ink is silver/silver chloride mixed particles and polyurethane resin (80%/20% *w/w*), with a viscosity of 70–180 Pa.s (JLL-20, Shanghai Julong Electronic Technology Co. Ltd., Shanghai, China). Before the printing, the ink was agitated at low speed for 10–15 min to make the ink mix well.

For the design of the electrode, the main considerations are the shape, size, and connecting wires. The common shapes of flexible ECG electrodes are square or round, and Krasteva et al. [29] found that the unevenness distribution of the current density of round electrodes is about 30% less than that of square electrodes. In general, within a certain range, the larger the contacted surface between the electrode and the skin, the lower the impedance and the more accurate the ECG signal acquisition [30]. Therefore, the ECG electrode was designed to be round with a diameter of 30 mm, and the connection part was a long rectangle (50 mm × 5 mm), which is easy to connect with the lead wire.

The screen-printing process (as illustrated in Figure 2) of the textile electrode was operated by a semi-automatic pneumatic cylindrical screen-printing machine (KM-SY3050A/V/T, Shenzhen Jincheng Kaimao Machinery Equipment Co., Ltd., Shenzhen, China.) with a velocity of 170 mm/s. The distance between fabric and screen frame was 4 mm, and the angle of the squeegee was 85° to the horizontal plane to ensure effective scraping of the conductive ink.

The labeled sheet resistance of this conductive ink was lower than 0.1 Ω/Sq with the recommended curing conditions of 100 °C and 20–30 min. All the textile electrodes were cured in a Precision Air Blast Drying Oven (BPG-9106A, Shanghai Yiheng Scientific Instrument Co., Ltd., Shanghai, China) at 100 °C for 30 min after screen printing.

For better illustration, the features of the printed textile electrode and the microscopic morphology were observed by the scanning electron microscope (SEM, ZEISS Gemini 300, Jena, Germany). Figure 3a is the partial enlarged boundary of the printing pattern (magnified 50 times); the left part is the primordial fabric, and the right part is the printed conductive lay. It can be concluded that the structure of the fabric is a regular satin fabric with smooth surface for easy screen printing. Due to the special texture of the fabric, the screen-printing layers are generally uniformly distributed, but there are slight ups and downs in the thickness direction. Figure 3b shows the cross-sectional morphology of the printed electrode with a magnification of 100 times, and the thickness of the printed conductive layer is continuous and complete, with a thickness of 31.54 ± 9.04 μm. The surface of the fabric is smooth and flat but uneven in height. This may be one of the reasons that the layer thickness of printing conductive ink is not uniform.

Figure 4 is the surface microscopy of the printed textile electrode with magnifications of 500 and 3000 times, respectively. The main components of printed conductive ink are silver and silver chloride, and the nanosilver is flaky to increase the electrical conductivity, while silver chloride is granular to ensure the electrochemical stability. More importantly, the Cl^−^ ion participates in free charge exchange, which prevents charge accumulation on the interface between the electrodes and skin [21]. From Figure 4, it can be concluded that the silver chloride particles are uniformly dispersed in the flaky nanosilver, guaranteeing excellent electrical and electrochemical properties of the printed electrodes. 

### 2.2. Electrical/Electrochemical Characterization

The cardiomyocytes contract and expand regularly to generate biologic potentials, which are transmitted to the surface of body and collected by the ECG electrodes. The electrode converts ionic potentials of tissue into a stream of electrons to form regular ECG signals. As the typical characteristics of ECG signals are low frequency (1~100 Hz), weak signal amplitude (mV scale), and large individual variation, the performance of the ECG electrode is very critical. Generally, the electrical and electrochemical properties of the ECG electrode are two very key indicators. The impedance of the ECG electrode is required to be low enough, and the electrochemical performance is required to be excellent and stable. In this paper, the resistance in direct current (DC) and electrochemical properties of the printed textile electrode were evaluated.

The DC resistance of the textile electrode was measured by using a two-point probe method with a digital multi-meter on the surface of the printed layers.

The electrochemical properties including open circuit voltage, cyclic voltammetry test, and electrochemical impedance spectroscopy were measured with an electrochemical workstation.

As shown in Figure 5, the electrochemical properties were tested in a three-electrode system with an electrochemical workstation (DH7000C, Jiangsu Donghua Analytical Instrument Co., Ltd., Taizhou, China). Considering the practical applications of the ECG electrodes, artificial sweat (ZW-HY-1000, Shenzhen Zhongwei Equipment Co. Ltd., Shenzhen, China) was used to replace the conventional sodium chloride (NaCl) solution. Artificial sweat is a mixed solution containing sodium chloride, lactic acid, urea, demineralized water, and so on. The pH of the artificial sweat is 6.5 [31], formulated according to ISO3160-2 standards [32]. 

In this experiment, a rectangular part (20 mm × 5 mm) of the textile electrode was cut off to test. About 1/2~2/3 of the sample was immersed into the electrolyte solution, while the entire platinum electrode was immersed into the electrolyte. The sample to be tested was set as the working electrode, the platinum electrode was set as the counter electrode, and the saturated calomel electrode was set as the reference electrode.

During the electrochemical measurement, in the open potential mode, the sampling time was set as 30 s, and the time interval was 0.5 s. In the cyclic voltammetry mode, the initial potential was set to −1 V, while the highest potential was 0.7 V, and the scan rate was set to 0.01 V/s. In the electrochemical impedance spectroscopy mode, the frequency was swept from 1000 Hz to 0.01 Hz.

### 2.3. Durability Test

To comprehensively evaluate the reliability of the printed textile electrodes, two groups of tests were carried out to evaluate the deformation of electrical property after specific bending and washing cycles.

The effect of mechanical performance was studied by carrying out a repetitive bending test with a radius of curvature of 14 mm. The resistance per unit length was recorded after each bending cycle using a two-point probe method. The electrical resistances before bending and after bending were measured, and the resistance change rate was calculated. The relative electrical resistance of the textile electrode over 5000 bending cycles was measured and compared.

Referring to the ISO 6300 [33] washing standard, the textile electrode was dipped and stirred in a thermostatic water bath with a 5 g/L soaping agent at 40 °C for 20 min and then dried at 60 °C in the air blast drying oven. The resistance per unit length was recorded after each washing cycle using a two-point probe method. The relative electrical resistance of the textile electrode over 20 washing and drying cycles was measured and compared.

### 2.4. ECG Monitoring Acquisition

During the ECG monitoring, Einthoven’s triangle theory was used to form a standard three-electrode system. As shown in Figure 6, the textile electrodes were fixed on the right hand, right leg, and left leg, which form a standard second lead (II). To investigate the signal acquisition capability of the textile ECG electrode, a healthy male (age: 21, height: 177 cm, weight: 52 kg) without history of myocardial infarction or ventricular hypertrophy was monitored by wearing the textile electrode (as shown in Figure 7). The ECG signals were collected by the ECG-PPG Data Acquisition Device (KT-905, Shanghai Pukang Co., Ltd., Shanghai, China). The sampling rate was 200 Hz in the continuous sampling mode. Without a filtering module, it can only collect and display the ECG signals on the screen of this device in real time and then upload the original data to the computer. That is to say, the collected original data was presented by MATLAB software in this paper without being amplified and filtered by this data acquisition device or processed by any signal processing software.

Furthermore, an ECG Signal Simulator (KS-300, Shanghai Pukang Co., Ltd., Shanghai, China) was connected to the ECG-PPG Data Acquisition Device to verify the entire experimental process. The output of this ECG Signal Simulator was set as normal sinus rhythm with a heart rate of 80 BPM and signal amplitude of 1 mV. It was connected to the ECG-PPG Data Acquisition Device with the standard second lead (II), and the output waveform was measured. For comparison, the electrocardiogram of a healthy adult was monitored by fixing the textile electrodes on the specific location to form the standard second lead (II).

## 3. Results and Discussion

### 3.1. Electrical and Electrochemical Analysis

The DC resistance of the textile electrode was measured to be 0.2 ± 0.005 Ω/cm. It indicated a good electrical conductivity of the cured printing layer, which helps in ECG signal acquisition.

In the electrochemical experiments, as shown in Figure 8a, the open circuit voltage was about 88 mV, and the fluctuation was less than 1 mV, much lower than that of silver-plated fabric electrodes [12]. The conductive ink in this work is silver/silver chloride; the silver provides excellent conductive pathways, and silver chloride provides stable electrochemical properties such as half-cell potential. This experiment demonstrates the good stability of the textile ECG electrode prepared in this work.

For the cyclic voltammetry characteristic (as shown in Figure 8b), the curve shows a duck shape with a closed loop. As can be seen from the curve, there are oxidation peaks and reduction peaks in the cyclic voltammetry characteristic curve and the area enclosed by the curve is small. This indicates that the electrode can undergo redox reactions, while the reversible reaction is poor. The larger the area included in the curve, the more likely the redox reactions are to occur. It has been shown that the area enclosed by the curve is positively correlated with the defibrillation recovery performance of cardiac electrodes [34]. In fact, the chloride ions react with silver ions in a redox reaction to produce silver chloride and electrons at the skin/electrode interface. That is to say, a large number of chloride ions on the skin surface or in sweat convert ionic electricity into electronic electricity when the electrodes are working. At the same time, the silver chloride in the electrode are decomposed into chloride and silver ions, forming a dynamic equilibrium.

In the electrochemical impedance spectroscopy mode (as shown in Figure 9a), the Bode curve indicates that the alternating current (AC) impedance of the electrode gradually decreases with the increase of frequency, and the impedance tends toward zero when it reaches a certain frequency. The impedance values of the dry electrode must be less than 2 kΩ, according to the Association for the Advancement of Medical Instrumentation (AAMI) [35]. As shown in the zoom-in picture of Figure 9a, the main frequency range of ECG signals is within 100 Hz, and the electrochemical impedance of the textile electrode in this work varies over a range of several hundred ohms within 100 Hz and obviously meets the above criteria.

During the signal acquisition, the electrode converts ionic potentials of tissue to a stream of electrons to form regular ECG signals. The completion of this process is reflected in the exchange of charged particles of the textile electrode and the electrolyte. The resistance during the process of this transfer exchange is called “transfer resistance” and corresponds to the diameter of the semicircular arc in the Nyquist diagram. It can be clearly distinguished from Figure 9b that the transfer resistance of the textile electrode is small, which is more conducive to the passage of physiological signals [36].

### 3.2. Durability Analysis

To evaluate the durability of the textile electrodes, the resistance after bending and washing for a number of times was evaluated. From Figure 10a, it can be deduced that the resistance almost remains the same after 300 bending cycles, and the resistance per unit length (centimeter) of the textile electrode increased from the initial 0.2 ohm to 0.5 ohm after 5000 bending cycle tests. Despite a 150% increase in relative resistance after 5000 bending cycles, the resistance per unit length was still no more than 0.5 ohm, exceeding the conductivity of many other fabric electronics [37]. Figure 10b shows the resistance variation of the textile electrode over 20 washing and drying cycles. It can be deduced that the resistance creeps up slowly and fluctuates slightly under washing conditions. After 20 washing and drying cycles, the resistance per unit length (centimeter) of the textile electrode increased from the initial 0.2 ohm to 0.38 ohm, increasing by 90% in relative resistance. The results of the durability evaluation experiments demonstrate that the textile electrode is able to maintain stable resistance under certain bending and washing effects.

In order to investigate the effect of the textile electrode under bending or washing cycles, the surface morphology of the original textile electrode, electrode bending for 5000 cycles, and electrode washing for 20 cycles were compared, and the SEM images of the local area are shown as Figure 11. 

By careful comparison, there is a visible crack near the bending position (as shown in Figure 11b), which may cause the partial or local discontinuity of the conductive path. After 20 washing and drying cycles, the holes or cracks increased insignificantly (as shown in Figure 11c. This may be because the washing process was dipped and stirred in a thermostatic water bath, not a washing machine with external mechanical force. Bending or washing cycles caused cracks and porous structures of the conductive layers, resulting in a decrease in the conductivity of the conductive layers. However, from the results of morphology comparison (Figure 11), the differences were not obvious before and after the mechanical damage, which indicates that the binding fastness of the conductive layer is very strong. It is probably the reason that the resistance remains almost unchanged before and after bending and washing cycles. All in all, the experimental results demonstrate good mechanical properties and durability of this textile electrode, contributing to the realization of sustainable wearable fabric electronics.

### 3.3. Signal Acquisition Analysis

Finally, the textile electrode was evaluated for the ability to acquire ECG signals. The textile electrodes were fixed on the specific location of a healthy adult to form the standard second lead (II), and the electrocardiographic signals were monitored while the object was lying flat on its back. The simulator was set as normal sinus rhythm with a heart rate of 80 BPM and signal amplitude of 1 mV. As illustrated in Figure 12, it is a significant and stable QRS waveform with normal sinus heart rate, and the maximum magnitude of a QRS-wave is 1.084 mV. By calculating the corresponding time of the peak of the ECG waveform, the number of heartbeats per unit of time can be simply calculated within a known sampling frequency. Here the sampling frequency is 200 Hz, and the time interval between two peak heartbeats is calculated to be 0.75 s. Therefore, the heart rate is 80, which is equivalent to the value set by the ECG signal simulator. It can be deduced that this ECG acquisition device can accurately acquire and display original ECG signals. After the whole method and process of the measurement were verified, the self-prepared textile electrode was utilized to collect ECG signals of humans.

Figure 13 shows the original data collected from the textile electrode. The acquired ECG signal contains a complete and stable cardiac cycle, including obvious P-wave, QRS-wave, and T-wave with normal sinus heart rate, and the maximum magnitude of the QRS-wave is 1.073 mV. With a sampling frequency of 200 Hz, the number of heartbeats per unit of time was simply calculated as 0.885 s, and the heartrate was calculated to be 68 BPM. This indicates that the textile electrodes can be used as ECG electrodes and that reliable ECG signals can be monitored. 

However, there is obvious jittering of the baseline ripple in the P-Q interval and Q-T interval of the waves compared with the signals of the ECG signal simulator. This should be caused by the large contact impedance at the interface of the electrode and the skin, since the contact conditions between the textile electrode and the skin are not as favorable as those of the medical Ag/AgCl electrode with conductive gel. According to the classic equivalent circuit model for an electrode-skin interface, the contact impedance is proportional to the resistance of the electrode(Rs) and affected by the double-layer capacitor between the electrode and skin(Cdl) and the charge transfer resistance of the electrode(Rct) [38,39]. As the printed textile electrode was fabricated with Ag/AgCl mixed conductive ink, the significant fluctuations of the ECG signals are not only caused by the conductivity lower than pure metal but also by the non-uniform distribution of silver and the silver chloride particles. Additionally, textile electrodes are dry electrodes, and the charge transfer resistance increases due to the lack of the skin hydration effect of gel, which also increased contact impedance. Another factor is that the textile electrodes are prone to slip during monitoring due to lack of immobilization by the conductive gel, resulting in more susceptibility to motion artefacts [40]. Furthermore, the EMG signals were caused by muscular activity [41] even though the subject was lying relaxed during the measurement. Those is the main reasons for the significant fluctuations of the baseline ripple compared to the original simulator in the ECG signals. 

After all, this is the original signal after acquisition, without filtering of electromyogram and industrial frequency interference. The textile electrode should be able to meet the requirements after effective filtering processing.

## 4. Conclusions

In conclusion, a textile electrode was fabricated by screen printing Ag/AgCl conductive ink on ordinary polyester fabric. A series of characterizations were carried out to comprehensively analyze the performance of the textile electrode, including surface morphology, electrical and electrochemical properties, durability, and capabilities of ECG signal acquisition. The results showed that the textile electrode has much lower resistance of 0.2 Ω/cm and can be maintained within 0.5 Ω/cm after 5000 bending cycles or 20 washing and drying cycles, indicating excellent flexibility and durability. The electrochemical properties of open circuit voltage, cyclic voltammetry, and electrochemical impedance spectroscopy were measured and evaluated, and the results demonstrated the good stability of the redox reactions. Furthermore, the ECG signals can be effectively acquired with this textile electrode, and the ECG waveforms are clear and complete with significant P-waves, QRS-waves, and T-waves. Therefore, the textile electrode fabricated here can pave ways for wearable bio-potential monitoring systems for the prevention of cardiovascular diseases.

## Figures and Tables

**Figure 1 polymers-15-03665-f001:**
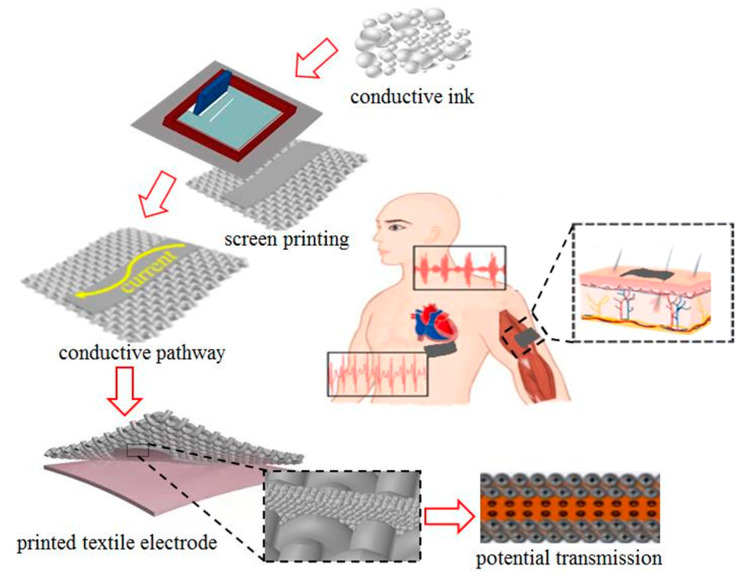
Diagram of ECG monitoring with textile electrodes.

**Figure 2 polymers-15-03665-f002:**
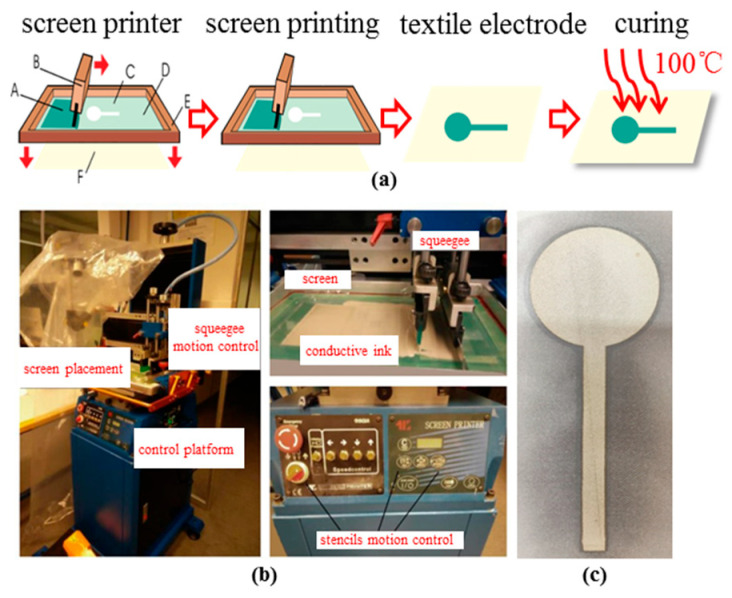
Preparation process of textile electrode. (**a**) Schematic diagram of screen-printing process (A-conductive ink, B-squeegee, C-screen, D-pattern to be printed, E-screen frame); (**b**) screen printing machine; (**c**) photo of the fabricated textile electrode.

**Figure 3 polymers-15-03665-f003:**
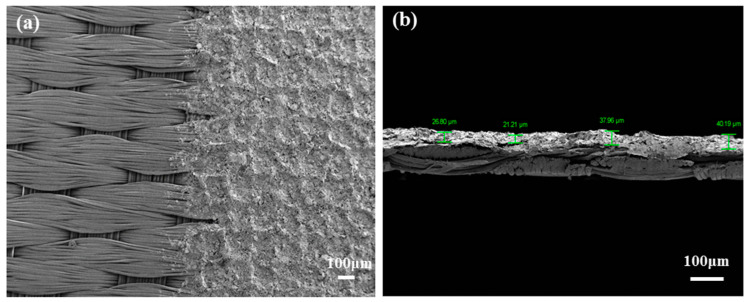
The SEM images of printed textile electrode: (**a**) boundary of the printing lay with a magnification of 50 times; (**b**) cross-sectional morphology of the printed lay with a magnification of 100 times.

**Figure 4 polymers-15-03665-f004:**
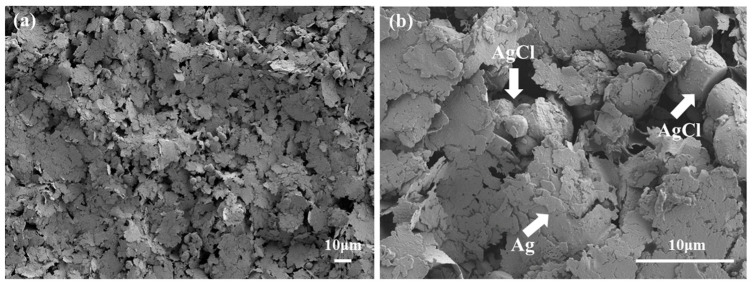
The surface SEM images of printed textile electrode (**a**) magnified by 500 times; (**b**) magnified by 3000 times.

**Figure 5 polymers-15-03665-f005:**
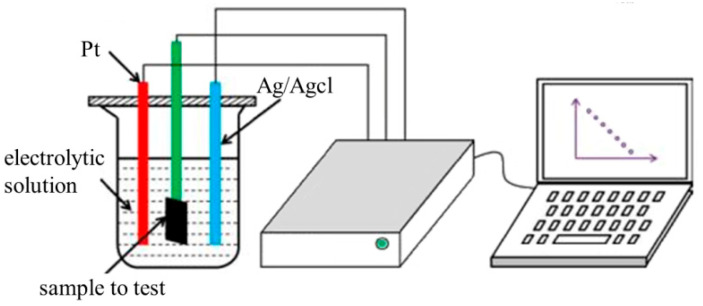
The schematic diagram of electrochemical performance test.

**Figure 6 polymers-15-03665-f006:**
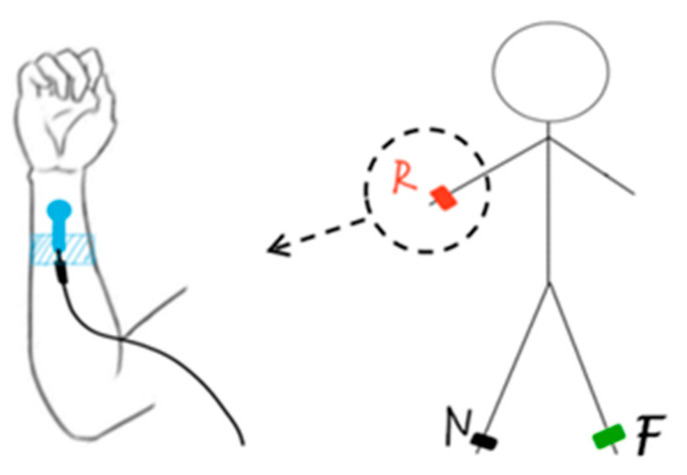
Standard second lead (II) ECG monitoring system.

**Figure 7 polymers-15-03665-f007:**
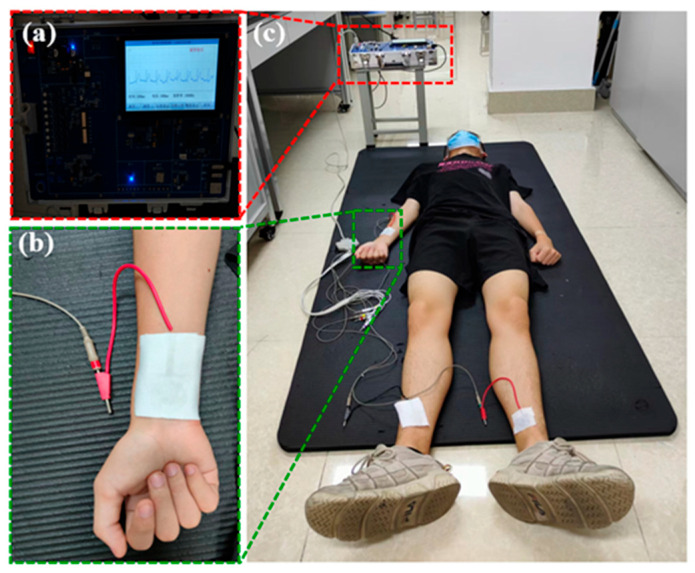
Photograph of ECG signal-acquisition process. (**a**) electrocardiogram waveforms displayed by an electrocardiograph, (**b**) textile electrode, (**c**) electrocardiographic monitoring for an adult.

**Figure 8 polymers-15-03665-f008:**
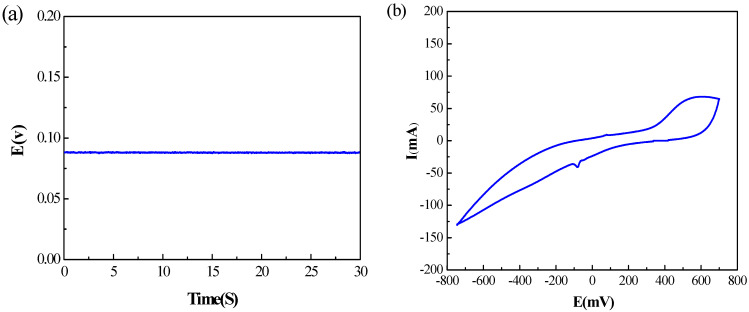
(**a**) Open circuit voltage; (**b**) cyclic voltammetry characteristics.

**Figure 9 polymers-15-03665-f009:**
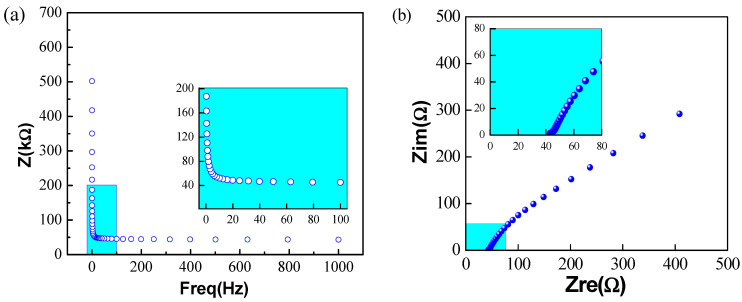
Electrochemical impedance spectroscopy mode: (**a**) Bode; (**b**) Nyquist.

**Figure 10 polymers-15-03665-f010:**
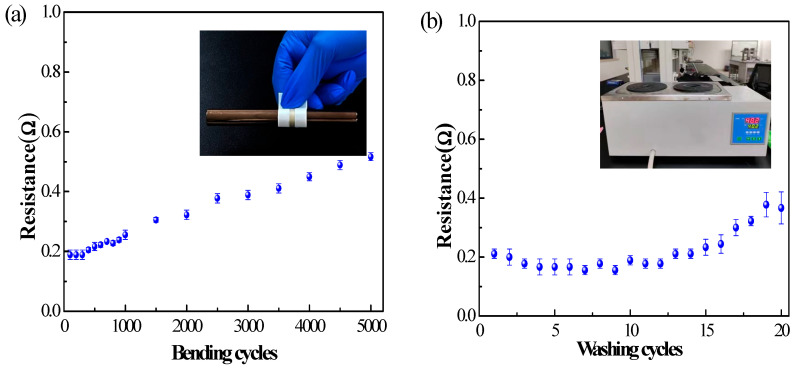
Resistance of the textile electrodes (**a**) during 5000 bending cycles; (**b**) during 20 washing cycles (thermostatic water bath).

**Figure 11 polymers-15-03665-f011:**
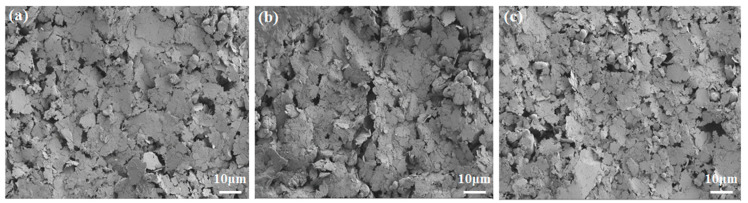
The surface SEM images of printed textile electrode (magnified by 1000 times): (**a**) the original electrode; (**b**) after bending for 5000 cycles; (**c**) after washing for 20 cycles.

**Figure 12 polymers-15-03665-f012:**
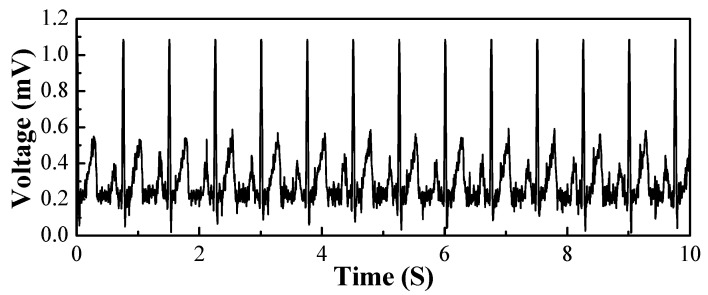
Original data collected from ECG Signal Simulator.

**Figure 13 polymers-15-03665-f013:**
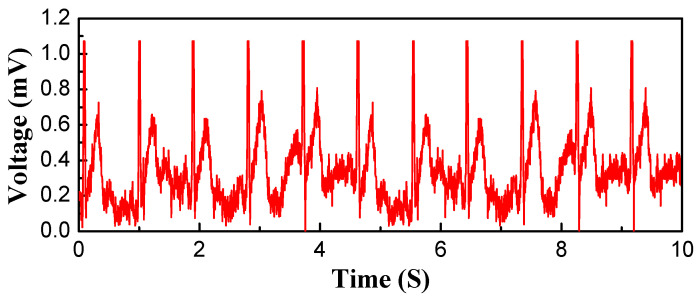
Original data collected from textile electrode.

## Data Availability

The authors confirm that the data supporting the findings of this study are available within the article.

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
