# Peer review of "Washable and Flexible Screen-Printed Ag/AgCl Electrode on Textiles for ECG Monitoring"

_polymers, 2023, doi:10.3390/polym15183665_

Round 1
Reviewer 1 Report
In this article, the authors demonstrated a dry electrode for ECG monitoring by screen printing with Ag/AgCl conductive ink. This display textile ECG electrodes can be bent 5000 times or washed 20 times, demonstrating good flexibility and durability. However, there are some shortcomings in this article. Therefore, I think the article can be published in “Polymers” after a major revision. I have detailed my suggestions below:
1. Considering practical applications, simulated sweat should be used as the electrolyte in electrochemical tests.
2. The authors should add mechanical property data for textile ECG electrodes.
3. The effect of the thickness of the Ag/AgCl conductive ink layer on the ECG detection signal.
4. On page 7, line 195, the authors' description of “As the main frequency range of ECG signals is within 100 Hz, the electrochemical impedance of the textile electrode in this work varies over a range of several hundred ohms thin 100 and obviously meets the above criteria.” but the value of the impedance within 100 Hz is not visible from Figure 8a.
5. The authors should explain the reason for the large difference between the resistance values of DC and EIS.
6. Fig. 11 should be Figure 11 on page 8, line 240.
The language of this article should be further modified.
Author Response
We gratefully thank the editor and all reviewers for their constructive remarks and useful suggestions, which have significantly raised the quality of the manuscript and enable us to improve the manuscript. Each suggested revision and comment brought forward by the reviewers was accurately incorporated and considered. Below, the comments of reviewers are answered one by one. The literatures given below the “Response” of each issue have been correctly cited in the revised manuscript, and adhered here for the convenience of reading.

Reviewer 2 Report
The manuscript describes the fabrication and evaluation of a textile-based electrode for electrocardiogram (ECG) monitoring. In response to the limitations of traditional gel electrodes, such as allergic reactions and shelf life, the authors have developed a screen-printed textile electrode using silver/silver chloride (Ag/AgCl) conductive ink on polyester fabric. The electrode is reported to exhibit excellent electrical and electrochemical stability, with a consistent resistance within 0.5 Ω/cm, even after 5000 bending cycles or 20 washing and drying cycles. The ECG signal acquisition performance of the electrode is noted to be reliable, making the electrode a promising candidate for wearable electrophysiological sensing platforms.
While the work carried out presents important contributions in the field of wearable biomedical devices, the reviewer recommends a major revision before the manuscript can be considered for publication. The novelty of the presented work appears limited, as the fabrication method and the material used (Ag/AgCl conductive ink) are commonly employed in the field. Additionally, the manuscript lacks sufficient experimental detail, which may hinder repeatability and validation of the reported results. The reviewer recommends that the authors carefully revise the manuscript and address the following issues before publication:
Major issues:
1. The manuscript, entitled “Washable and Flexible Screen Printed Ag/AgCl Electrode on Textiles for ECG Monitoring,” bears significant resemblance to an existing publication titled “Washable and flexible screen printed graphene electrode on textiles for wearable healthcare monitoring.” The lack of acknowledgment and comparison with this prior work raises concerns about potential academic misconduct. To maintain the integrity of scientific research, it is essential to provide proper credit and references to preceding studies, especially when there is considerable overlap in research content and methodology.
2. At the 49th line on page 2, the authors assert the importance of durability evaluation under washing conditions. However, no supporting evidence or references are provided to validate this claim. The authors are advised to substantiate this assertion by citing relevant literature that emphasizes the significance of wash durability in textile electrodes.
3. Figure 8a illustrates the electrochemical impedance of the electrode across a broad frequency range (0 Hz to 1 GHz). Given that the intended application of the electrode is ECG signal acquisition, which occurs within a frequency range of 100 Hz, the display of impedance across the current frequency range appears excessive. To enhance clarity and relevance, the authors are advised to include a zoomed-in version of the figure, highlighting the impedance values in the low-frequency range pertinent to ECG signals.
4. In Figure 10c, it is challenging to discern whether the number of holes in the electrode increases or not. A quantitative assessment of this parameter would enhance the clarity of these results, providing definitive evidence of any change in the hole count.
5. Figure 12 compares the ECG data collected from the textile electrode with that from an ECG signal simulator. Notably, the data from the textile electrode display significant fluctuations relative to the original simulator data. A thorough discussion exploring the possible reasons for this disparity would contribute greatly to understanding the textile electrode's performance and potential limitations.
Minor issues:
1. There is a typographical error on page 2, line 45, where "AgCl" is incorrectly written as "Agcl".
2. On page 2, line 45, the abbreviation "PEDOT:PSS" is used without a preceding definition. Please include a full form for the readers' understanding.
3. On page 3, line 93, there is a typographical error in the resistance value "0.1 Ω/â–¡".
4. On page 7, line 195, there seems to be a typographical error in the term "AMMI".
5. On page 7, line 197, the numerical value "100" appears to be a typographical error.
Overall, the manuscript is well-written, with a clear structure and logical flow. However, there are several instances of typographical errors, and some terms and abbreviations are used without being defined. In addition, there are a few grammatical errors and awkward phrases that should be corrected.
Author Response

(The authors gave the same response as above.)

Round 2
Reviewer 1 Report
Accept
Reviewer 2 Report
All major issues have been fixed.